# Underwater light environment in Arctic fjords

Robert W. Schlegel[1], Rakesh Kumar Singh[2,3], Bernard Gentili[1], Simon Bélanger[2], Laura Castro de la Guardia[4], Dorte Krause-Jensen[5], Cale A. Miller[6], Mikael Sejr[5,7], Jean-Pierre Gattuso[1,8]

[1]Laboratoire d'Océanographie de Villefranche, Sorbonne University, CNRS, Villefranche-sur-mer, France
[2]Département de Biologie, Chimie et Géographie, Université du Québec à Rimouski, Rimouski, QC, Canada
[3]Centre for Remote Imaging, Sensing and Processing (CRISP), National University of Singapore, Singapore
[4]Norwegian Polar Institute, Fram Centre, Tromsø, Norway
[5]Department of Ecoscience, Aarhus University, Aarhus C, Denmark
[6]Department of Earth Sciences, Utrecht University, Utrecht, The Netherlands
[7]Arctic Research Centre, Aarhus University, Aarhus C, Denmark
[8]Institute for Sustainable Development and International Relations (IDDRI-Sciences Po), Paris, France

*Correspondence to*: Robert W. Schlegel (robert.schlegel@imev-mer.fr) and Jean-Pierre Gattuso (jean-pierre.gattuso@imev-mer.fr)

**Abstract.** Most inhabitants of the Arctic live near the coastline, including fjord systems where socio-ecological coupling with coastal communities is dominant. It is therefore critically important that the key aspects of Arctic fjords be measured as well as possible. Much work has been done to monitor temperature and salinity, but an in-depth knowledge of the light environment throughout Arctic fjords is lacking. This is particularly problematic knowing the importance of light for benthic ecosystem engineers such as macroalgae, which also play a major role in ecosystem function. Here we document the creation and implementation of a high resolution (~50-150 m) gridded dataset for surface photosynthetically available radiation (PAR), diffuse attenuation of PAR through the water column ($K_{PAR}$), and PAR available at the seafloor (bottom PAR) for seven Arctic fjords distributed throughout Svalbard, Greenland, and Norway, during the period 2003-2022. In addition to bottom PAR being available at a monthly resolution over this time period, all variables are available as a global average, annual averages, and monthly climatologies. Throughout most Arctic fjords, the interannual variability of monthly bottom PAR is too large to determine any long term trends. However, in some fjords, bottom PAR has increased in spring and autumn, and decreased in summer. While a full investigation into these causes is beyond the scope of the description of the dataset presented here, it is hypothesised that this shift is due to a decrease in seasonal ice cover (i.e. enhanced surface PAR) in the shoulder seasons, and an increase in coastal runoff (i.e. increased turbidity/decreased surface PAR) in summer. A demonstration of the usability of the dataset is given by showing how it can be combined with known PAR requirements of macroalgae to track the change in time of the potential distribution area for macroalgal habitats within fjords.

The dataset (Gentili et al., 2023a) is available on PANGAEA at: https://doi.org/10.1594/PANGAEA.962895

A toolbox for download and working with this dataset is available in the form of the FjordLight R package, which is available via CRAN (Gentili et al., 2023b), or may be installed via GitHub: https://face-it-project.github.io/FjordLight (last access: 8 December 2023).

## 1 Introduction

The Arctic Ocean is surrounded by three continents whose extensive coastlines ensure that coastal ecosystems are an important component of the overall Arctic marine realm. The area shallow enough for light to reach the seafloor is estimated to be approximately 3 million km2 (Gattuso et al., 2006), which is roughly equivalent to the central Arctic Ocean (3.3 million km2, PAME, 2016). Fjords are one of seven distinct coastscapes found in the Arctic and are common in Norway, Greenland, Iceland, and Eastern Canada (CAFF, 2019). Fjords are defined in a geographic context as deep narrow inlets of water, sometimes with a sill, a physical barrier that creates inner and outer deep areas, and are generally surrounded by steeply rising topography. Coupled with a high northern latitude location, this has historically meant that most Arctic fjord systems are strongly influenced by glaciers in a number of important ways. Due in part to the confluence of geography and cryosphere, Arctic fjord ecosystems are an order of magnitude more productive than terrestrial Arctic ecosystems, providing suitable areas for spawning grounds and nurseries of marine fauna (e.g. Spotowitz et al., 2022), acting as carbon sinks (Smith et al., 2015), and may even be productive enough for aquaculture development (Hermansen & Troell, 2012; Aanesen & Mikkelsen, 2020).

The light available throughout the water column referred to in this study is specifically limited to Photosynthetically Available Radiation (PAR). This is solar radiation found between the wavelengths of 400 and 700 nm and can be absorbed by the dominant photosynthetic pigments in marine primary producers (Morel, 1978). PAR diminishes as it penetrates the water column due to its optical properties. This reduction in the availability of PAR with depth can be estimated using the diffuse downwelling attenuation coefficient for PAR ($K_{PAR}$) of the water column. The higher the scattering (e.g. due to high/large sediment load and phytoplankton) and absorption (e.g. due to high concentrations of dissolved organic matter, organic detritus, minerals, and phytoplankton) in the water column, the higher the $K_{PAR}$. This is an important consideration as the PAR reaching the seafloor (PAR bottom or $PAR_B$) is one of the major limiting factors for the distribution, production, and composition of benthic phototrophic communities. The geographical distribution of PAR and $K_{PAR}$ therefore plays an important role in regulating the global carbon cycle through the control of light availability on the depth distribution of benthic primary producers (Gattuso et al., 2020).

In general, there are three known processes that affect the penetration of light through the water column in most Arctic coastal ecosystems, and particularly in Arctic fjords:

1) Loss of sea-ice has resulted and will continue to result in longer periods of open water, allowing greater penetration of light below the sea surface (Pavlov et al., 2019)
2) Suspended particles in the water column that originate from glacial or terrestrial run-off, or resuspension from increased fetch and wave action, limit light penetration (Frigstad et al., 2020; Nowak et al., 2021)
3) Cloudiness may increase as the Arctic warms, reducing incident PAR over the sea surface (Bélanger et al., 2013; Laliberté et al., 2021)



The processes listed above are likely to exhibit considerable regional and local variability, making it complex to
quantify trends in coastal PAR. This means that the drivers of light availability in fjords might follow different
trajectories in different geographical settings. While it has been well-established and quantified how light
availability and pelagic productivity have increased in the open Arctic Ocean due to reductions in sea-ice cover
(Pavlov et al., 2019), the response of benthic primary producers in fjords remains poorly constrained.


Benthic primary producers in Arctic fjords include microalgae (i.e., microphytobenthos), macroalgae (e.g. kelps
and encrusting corallines) and seagrass (Zostera marina, known as eelgrass, is the only seagrass that extends into
the Arctic zone). Kelps and seagrasses are canopy-forming and act as ecosystem engineers by creating vertical
structures used by a wide range of species, thereby supporting marine biodiversity (Wernberg et al., 2019). Even

in the Arctic, the areal extent, and production of kelps and other macrophytes can be substantial (Krause-Jensen
et al., 2020; Filbee-Dexter et al., 2022; Castro de la Guardia et al., 2023).

Due to light limitation, the highest abundance of benthic primary producers is restricted to narrow coastal
margins, with seaweeds dominating rocky shores, while rooted macrophytes and microalgae colonise sandy or

soft sediments. These macrophytes are well adapted to low light environments and tend to have low
compensating and saturating irradiances (e.g. Borum et al., 2002). It is therefore an important finding that the
biomass of these coastal communities has increased (Kędra et al., 2010; Bartsch et al., 2016). A possible regime
shift occurred in 1995 in the rocky-bottom community of a well-studied Arctic fjord (Kongsfjorden; Kortsch et
al., 2012). An Arctic-wide study showed a general increase in macroalgae abundance, productivity, and/or

biodiversity, accompanied by a poleward migration rate of 18-23 km per decade (Krause-Jensen et al., 2020).

In addition, the depth at which macroalgal biomass is highest in at least one Arctic fjord (i.e. Kongsfjorden) is
becoming shallower (Bartsch et al. 2016). The two main hypotheses for why macroalgal biomass is shifting to
shallower depths in some fjords are both related to PAR availability:

100       1) Less sea-ice cover means both less ice scour and more light penetration at a shallower depths, which is
preferred by macroalgae (Bartsch et al. 2016; Fredriksen et al., 2019 and citations therein; Wiktor et al.,
2022)
          2) Increasing turbidity (e.g. melting glaciers, increased wave action, and coastal erosion) inhibits light
penetration to the deeper depths where macroalgae have historically been found (Bartsch et al., 2016)


The shift towards darker water is known as water "darkening" or "browning", and has been documented at high
northern latitudes (Finstad et al., 2016), including in most fjords of Western Svalbard (1935-2007; Konik et al.,
2021) and mainland Norway (1935-2007; Aksnes et al., 2009). However, this trend is complex and spatially
variable. For example, in a given fjord, underwater PAR may be decreasing, but the areas furthest away from

the points of freshwater input may show an increase in PAR because they are less affected by sediment input.
Therefore, the same trend of increasing PAR observed in the open ocean due to sea-ice loss (Arrigo & van
Dijken, 2011) may also apply to these outer-fjord regions.



As the Arctic climate continues to change rapidly, it is predicted that light availability in the open ocean will
continue to increase due to sea-ice loss (Pavlov et al., 2019). This, combined with increases in temperature and
possibly also nutrient availability, would hypothetically be beneficial for some macroalgae (Goldsmit et al.,
2021; Assis et al., 2022), although too much heat could eventually become problematic (Filbee-Dexter et al.,
2016; Bass et al., 2023). Indeed, it has been shown that the depth distribution of macroalgae may increase with
increasing number of open water days (Castro de la Guardia et al., 2023). However, it is still very uncertain to
what extent, and for how long, PAR will continue to change in Arctic fjords (Walch et al., 2022). Dissolved
organic matter, which affects PAR availability, may also be altering benthic ecology (Sejr et al., 2022) or
otherwise negatively affecting macroalgae communities (Niedzwiedz & Bischof, 2023). Understanding these
changes is important and timely, as shallow Arctic fjord communities are predicted to shift from invertebrate-
dominated to algal-dominated communities (Kortsch et al., 2012; Lebrun et al., 2022).


The importance of underwater light for the distribution of benthic primary producers is undeniable, but there are
still many uncertainties about the overall spatial distribution and trends of PAR in Arctic fjords at the surface
and at depth. This is due to the fact that in situ PAR measurements are rare and  spatially sparse. To this end, we
document here the use of optical remote sensing data in combination with high resolution bathymetric maps to
estimate PAR at the surface of the ocean, its water column attenuation ($K_{PAR}$), and combine them to estimate
PAR reaching the seafloor, $PAR_B$. We provide spatial summaries of PAR as well as time series showing how
PAR may have changed in the shallow zones (depth ≤ 50 m) from 2003-2022. Finally, we compare the present
state of $PAR_B$ with the known light requirements of key benthic macrophyte primary producers to highlight the
utility of this dataset.

**2 Methods**

**2.1 Study sites**

Arctic fjords share a common glacial origin and history, but there are many differences between them, including
latitude, climate, bathymetry, freshwater input, orientation, and seasonal ice cover. The study sites for this data
product were chosen in order to include an appropriate range of environments within the area considered as the
European Arctic (25°W - 60°E and 66°N - 90 °N), subdivided here into mainland Norway, Svalbard, and
Greenland (Table 1, Fig. 1).

**Table 1:** Study sites included in this dataset, with summary notes on their state of glaciation and seasonal sea-ice cover.
Latitude values are approximated from the middle of the fjord system and are provided here as a general indication.

| EU Arctic sector | Fjord name | Latitude | Glaciation | Sea-ice cover |
|---|---|---|---|---|
| Norway (north) | Porsangerfjorden | 70.5°N | Lost glaciers and ice a long time ago | Lost sea-ice cover a long time ago |
| Svalbard (west) | Kongsfjorden, | 79, 78.5°N | Advanced stages of glacier retreat | Recent loss of sea-ice cover |





|  | Isfjorden |  |  |  |
|---|---|---|---|---|
| Svalbard (east) | Storfjorden | 78°N | No measurable glacier retreat | Seasonal sea-ice cover with no measurable sea-ice loss. |
| Greenland (east) | Young Sound | 74.5°N | Perhaps in the early stages of glacier retreat | Seasonal sea-ice cover but entering an early stage of sea-ice loss. |
| Greenland (west) | Qeqertarsuup Tunua, Nuup Kangerlua | 69, 64.5°N | Middle stages of glacier retreat | Seasonal sea-ice cover, but entering an advanced stage of sea-ice loss. |


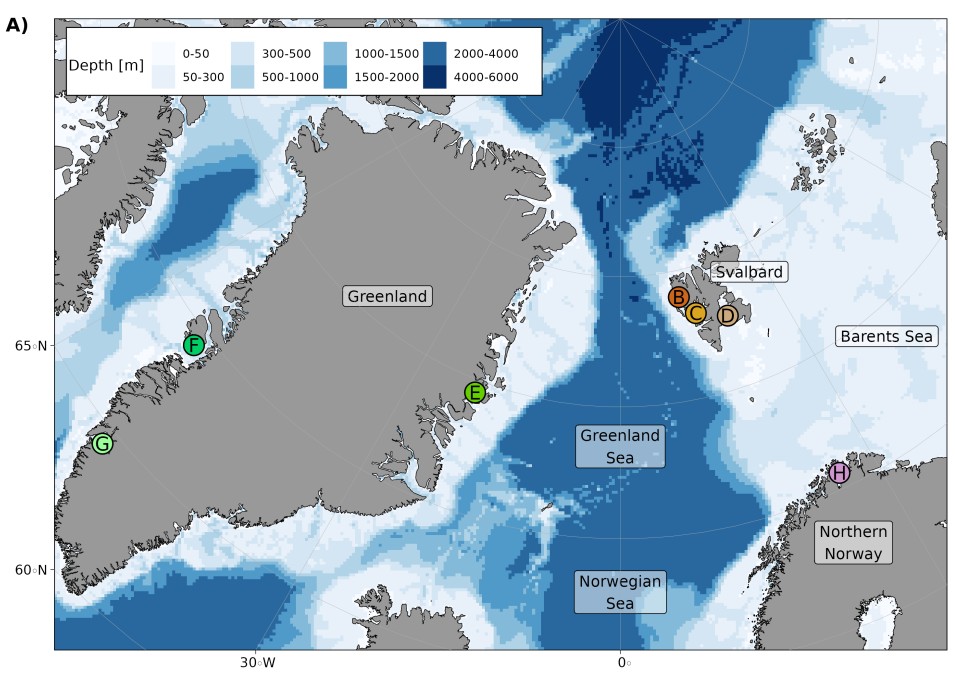

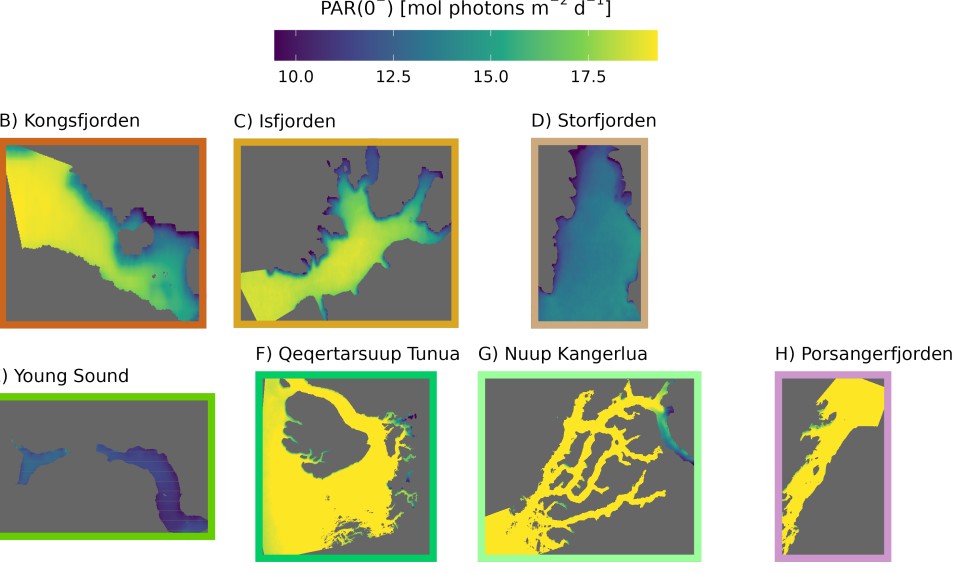

**Figure 1:** The location (denoted by dots in the upper panel) and regional subsets of the global PAR just below the water surface (PAR(0⁻); bottom panels) for the seven sites included in the FjordLight dataset. The scales for PAR(0⁻) (mol photons m-2 d-1) are the same in all panels, but the scales for longitude and latitude differ. Note that differences in PAR(0⁻) between sites are generally due to the difference in the seasonal cycle of sea-ice cover. The colours of the frames in panels B-H correspond to the colours of the dots in panel A.



The fjords of Northern Norway presently lack glaciers (e.g., Porsangerfjorden; Table 1). Thus freshwater inputs
are limited to terrestrial and riverine runoff, which may cause more darkening/browning of waters than glacial
runoff. These fjords also have little to no sea-ice cover throughout the entire year, making them systems where
surface PAR is more available in the spring and autumn relative to other Arctic regions. These fjords may be
precursors of fjords presently associated with glaciers elsewhere in the Arctic.

Western Svalbard fjords (e.g. Kongsfjorden and Isfjorden) are currently undergoing radical changes primarily
driven by a warming climate (Table 1). These fjords are experiencing the effects of rapidly melting glaciers and
drastic declines in sea-ice cover. Increased runoff has also led to a darkening of the nearshore waters, making
the changes in $PAR_B$ in this region unclear. Eastern Svalbard fjords (e.g., Storfjorden) are not yet heavily
impacted by climate change, but could be relatively soon. They still have relatively stable glaciers and seasonal
sea-ice cover. Therefore, seasonal surface PAR values in these fjords are not expected to differ significantly
from the historically stable baseline.

The selected Greenlandic fjords in the east (Young Sound) and west (Qeqertarsuup Tunua, Nuup Kangerlua)
show different degrees of glaciation and sea-ice cover (Table 1). Nuup Kangerlua is sea-ice-free year round
while Young Sound has a short sea-ice-free period of 2-3 months per year. The effects of climate warming (e.g.
glacier retreat) are greater in the west than in the east, but overall less than in the rest of the EU Arctic. This
means that the seasonal cryosphere cycle shows fewer signs of breakdown due to climate change, as has been
observed in Northern Norway, and is currently happening in Western Svalbard.

In addition to the changes in sea-ice cover/glaciation in Arctic fjords, latitude introduces a strong gradient in the
seasonal light regime (with increasing length of polar day and night towards the north), which plays an
important role in shaping the light climate of Arctic fjord ecosystems.

**2.2 Data Sources**

**2.2.1 Bathymetry data**

In order to accurately calculate $PAR_B$, it was necessary to utilise the highest resolution bathymetric data
available. The base layer used for all sites was v4.2 of the International Bathymetric Chart of the Arctic Ocean
(IBCAO; Jakobsson et al., 2020), an Arctic specific data product produced by the General Bathymetric Chart of
the Oceans (GEBCO). IBCAO contains all bathymetric data from 64°N to the pole at a gridded resolution of
m on the IBCAO Polar Stereographic projection (WGS 84; EPSG:3996). However, higher resolutions were
185 often available within the focal study sites for this dataset, and these were used wherever possible.

For Northern Norway (Porsangerfjorden) and Svalbard (Kongsfjorden, Isfjorden, and Storfjorden), data with a
50 m resolution were available from the Norwegian mapping authority
(https://dybdedata.kartverket.no/DybdedataInnsyn/). Gaps within the bathymetry of the Svalbard sites from this
source were filled with IBCAO data, and interpolated down to 50 m. For Western Greenland (Qeqertarsuup



Tunua and Nuup Kangerlua), 150 m data were utilised from v5.0 of the IceBridge BedMachine Greenland
product (IDBMG4; Morlighem et al., 2017), which is on the NSIDC Sea Ice Polar Stereographic North
projection (WGS 84; EPSG:3413). For Eastern Greenland (Young Sound), a site-specific dataset created by
Rysgaard et al. (2003) and subsequently improved with additional data was used. This has a spatial resolution of
100 m on a WGS 84 datum with EPSG:4326 projection.

To match with the gridded satellite data (see 2.2.2), all bathymetric data were re-interpolated from their native
projection was to the even grid cell system of the Standard Global Degree Decimal Projection (WGS 84;
EPSG:4326) using the highest resolutions mentioned above.

**2.2.2 Satellite data**

The top-of-atmosphere (TOA) radiance MODIS-Aqua Level-1A (L1A) data were acquired from NASA's Ocean
Biology Distributed Active Archive Center (OB.DAAC; https://oceancolor.gsfc.nasa.gov), covering the study
area from January 2003 to December 2022. The L1A data were processed to Level-2 at the native resolution of
MODIS ocean colour bands (~1 km) using SeaDAS v8. The atmospheric correction algorithm in SeaDAS was
modified to use the aerosol correction of Singh et al. (2019), which has been shown to improve the accuracy of
retrieving water-leaving radiance, particularly in turbid coastal waters.

In addition to the ocean colour data from MODIS-Aqua, the Earth Probe (EP) Total Ozone Mapping
Spectrometer (TOMS; TOMS Science Team, 1998) and the Ozone Monitoring Instrument (OMI; Bhartia, 2012)
onboard Aura were used to obtain ozone optical thickness and near real-time sea-ice concentration using passive
microwave radars was obtained from National Snow and Ice Data Center (NSIDC; Maslanik & Stroeve, 1999;
Meier et al., 2021).

**2.3 Analysis of remote sensing images**

The PAR just below the water surface (PAR(0⁻)) was calculated following the radiative transfer-based approach
of Singh et al. (2022). This method has been found to work adequately at high solar zenith angles, which is the
usual case for satellite-acquired optical signals in the Arctic region. At high latitudes, the importance of using
PAR(0⁻) rather than  PAR above the surface (PAR(0⁺)) becomes more evident, as the higher solar zenith angle
results in a significant difference between the PAR reaching the water surface and the PAR entering the water
column (Gregg & Carder, 1990). In polar regions, the daily average solar zenith angle is mostly higher than 55°
(Hartmann, 2016). This algorithm is integrated with a per-pixel flagging approach to differentiate between open-
water, sea-ice, and cloud, which increases the robustness of model inputs for calculating the sub-surface PAR in
ice-covered waters.

The atmospheric parameters computed from MODIS-Aqua data are utilised to compute cloud optical thickness,
while atmospherically corrected products are used to compute the ice-cloud-water flag and surface albedo for
the PAR(0⁻) (Singh et al., 2022; for details about the lookup tables used to compute PAR(0⁻), see Laliberté et al.,



2016). The ozone optical thickness was acquired from TOMS and OMI (section 2.2.2). In addition, sea-ice concentration from NSIDC was used to compute the surface albedo under the clouds. With these inputs and the solar zenith angle, the daily PAR(0⁻) was computed for each pixel and at a spatial resolution of ~1 km.


The PAR that penetrates the water column diminishes as it travels downwards due to scattering and absorption. This loss of PAR in the water column is governed by the attenuation coefficient for PAR ($K_{PAR}$), which is a function of the inherent optical properties (IOPs; absorption and backscattering coefficients) of the water column and the solar zenith angle. Hence, $K_{PAR}$, can be used to account for the attenuation of PAR in the water

column. Saulquin et al. (2013) found that the attenuation coefficient of downwelling irradiance at 490 nm (Kd(490 nm)) computed using IOPs estimated with a quasi-analytical algorithm (QAA; Lee et al., 2002, 2005) can be used to derive $K_{PAR}$ in coastal and turbid coastal waters with reasonable accuracy. Therefore, the $K_{PAR}$ values provided in this dataset were calculated using the remote sensing reflectance at 555 nm (Rrs(555)) from MODIS-Aqua using Saulquin et al. (2013) with the updated formulation of the QAA (Lee et al., 2013).

**2.4 $PAR_B$ calculation**

As mentioned in the previous section, PAR(0⁻) rather than PAR(0⁺) was used to compute $PAR_B$. PAR(0⁻) values were calculated using the SBDART (Santa Barbara DISORT Atmospheric Radiative Transfer) lookup tables described in Laliberté et al. (2016) and (Singh et al., 2022).

Using the Beer-Lambert Law, $PAR_B$ can be approximated as a function of PAR(0⁻) and $K_{PAR}$ for a known depth (m):

**EQ1:** $PAR_B = PAR(0^-) \times \exp(-K_{PAR} \times \text{bottom\_depth})$

Note that this equation can also be used to calculate PAR at any depth in the water column by replacing *bottom_depth* with the desired depth value in metres.

**2.5 *P*-functions**

We define a "*P*-function" as the percentage of the surface area in a shallow (depth ≤ 50 m) or coastal (depth ≤ 200 m) zone that receives $PAR_B$ greater than a threshold. The *P*-function was introduced by (Gattuso et al.,

2006, 2020) and can be calculated for a given region (i.e. a fjord) over a given time interval. Within this dataset the time periods available are: global (i.e. the full 20 years of data; 2003-2022), yearly (i.e. a year from 2003 to 2022), or the climatology for a given month (i.e. March to October - averaged over the full 20 years of data).

While a more detailed explanation may be found in section 2.5.2 of Gattuso et al. (2020), it is relevant to the

dataset being presented here to see how the data for the P-functions were calculated. Let *E* be a value of irradiance (expressed in mol photons $m^{-2}$ $d^{-1}$) and d a given day. For this day, let $S_{a,d}$ be the available surface (i.e. the total surface of pixels for which an irradiance value is available), and $s_d(E)$ the total surface of pixels



collecting irradiance greater than $E$. The $P$-function for a given time interval of n days $I = \{d_1, d_2, ..., d_n\}$ is therefore:


**EQ2:** $P_I(E) = 100 \sum_{i=1}^{n} s_{d_i}(E) / \sum_{i=1}^{n} S_{a,d_i}$

We may apply this by letting $P$ be a $P$-function and $S_{geo}$ the surface of the shallow coastal area of the fjord (0-50 m), the area receiving $PAR_B$ above a given threshold (expressed as $s(E)$ and measured in mol photons $m^{-2}$ $d^{-1}$)

is:

**EQ3:** $s(E) = S_{geo} \dfrac{P(E)}{100}$

The threshold value assigned to $E$ in eq. 3 could be a given benthic light requirement based on field

observations, as presented in the next section for a number of Arctic macroalgae.

**2.6 Benthic light requirements**

As an exercise to demonstrate the usability of the new dataset, an analysis of the light requirements of benthic macroalgae (kelps) was performed. This required a literature review of the light requirements and depth extensions of these organisms. It was found that the minimum light requirements ($E_{min}$) of Arctic kelps are

typically between 40 and 50 mol photons $m^{-2}$ $y^{-1}$, often equivalent to about 1% of surface irradiance (Table 2). The depth ranges of these organisms vary, but within fjords most are found between 0 and 20 m depth.

**Table 2:** Minimum light requirement ($E_{min}$; mol photons $m^{-2}$ $y^{-1}$) of Arctic kelps, the corresponding percentage of surface irradiance (S.I), the corresponding depth limit, and the species considered. A: *Alaria* (*esculenta*) or *Agarum* (*clathratum*), S:
*Saccharina*, L: *Laminaria*.

| Region | Latitude | $E_{min}$ | S.I. | Depth limit | Species | Reference |
|---|---|---|---|---|---|---|
|  | °N | mol photons $m^{-2}$ $y^{-1}$ | % | m |  |  |
| *Svalbard* |  |  |  |  |  |  |
| Hansneset, Kongsfjorden | 78.98 | 42 | — | 15 | *A. esculenta* | Bartsch et al. (2016) |
| *Greenland* |  |  |  |  |  |  |
| Young Sound | 74 | 40 | 0.7 | 15-20* | *S. latissima* | Borum et al. (2002) |





| | | | | | | |
|---|---|---|---|---|---|---|
| Disko Bay | 67 - 70 | — | Slightly >1 | ca. 60 (max) | *A. clathratum* (typically), *S. latissima, L. solidongula* | Krause-Jensen et al. (2019) |
| ***Iceland*** | | | | | | |
| Various sites | 65.3 - 65.85 | 34, 102** | 0.6-1.9 | Down to 27 m | *L. digitata, L. hyperborea* | Gunnarsson (1991) |
| ***Canada*** | | | | | | |
| Southampton Island, Nunavut | 62 - 67 | 49 | 1.4 | 37 (median) | Mix*** | Castro de la Guardia et al. (2023) |
| Igloolik Island, Foxe Basin, Nunavut | 69.4 | 49 | — | 20 | *L. solidongula* | Chapman & Lindley (1980) |
| ***Alaska*** | | | | | | |
| Stefansson Sound, Beaufort Sea | 70.3 | 45-50 | Down to 0.2 | 5 | *L. solidongula* | Dunton (1990) |
| **Arctic** | | | | | | |
| *Median across sites* | *68.2* | *47* | *0.85* | *20* | *Mix* | *—* |

*Young specimens with thin thalli extended to 20 m, while older specimens with thicker thalli and poorer light utilisation capacity extended solely to 15 m.

** Each value provided here corresponds to the species listed in the same row.

***Depth limits were reported for the kelp assemblage in general, comprising high-canopy kelps including *S. latissima* (var. hollow buoyant stipes), *A. esculenta, L. solidungula*, and low canopy kelps including *A. clathratum* and kelp juveniles, with the low-canopy assemblage often forming the depth limit.

## 3 Results

### 3.1 PAR(0⁻) and $K_{PAR}$

PAR(0⁻) and $K_{PAR}$ are available for all sites as global mean, annual mean, and monthly climatological mean values. By taking the median (spatial) value for each site for pixels in the shallow zone (i.e. pixels with depth ≤ 50 m) we can better visualise the seasonal changes (Figure 2).



**Figure 2:** Median monthly climatology values for A) PAR just below the surface (PAR(0⁻)), and B) attenuation coefficient of PAR ($K_{PAR}$) for the shallow zone pixels (depth ≤ 50 m) from each site. Note the seasonal cycle in PAR(0⁻) for all sites, but the different patterns for $K_{PAR}$.




There is a clear seasonal cycle in the monthly climatology of light penetrating the surface of the shallow zone at all seven study sites (Fig. 2A). The median value of PAR($0^-$) across all sites starts relatively low in March, where it then increases to a peak sometime between June and July, before decreasing again until September to October. The delay in seasonal PAR($0^-$) peak relative to solar solstice is due to the sea-ice cover, unlike the

PAR($0^+$) (i.e. PAR above the surface) that usually occurs prior to the solstice (Laliberté et al., 2021).

The monthly climatology of the shallow zone $K_{PAR}$ (Fig. 2B) shows two different patterns. The first pattern, found in all Svalbard and Eastern Greenland sites, is a stable or decreasing $K_{PAR}$ from March to June, and then an increase until June to September, before decreasing again until the end of the illuminated part of the year.

The second pattern, which is found in Western Greenland and Northern Norway, is a stable or fluctuating $K_{PAR}$ until August, followed by a rapid increase up until the end of the illuminated part of the year. While it is beyond the scope of the description of this dataset to investigate these patterns in detail, it is hypothesised here that the first pattern is representative of a system that is still dominated by a marine terminating glacier, and that the bulk of the turbidity in the water (i.e. $K_{PAR}$) is due to the glacial runoff during the warmest summer months (July,

August). This is why it starts to build up in June, but decreases after a couple of months. The second pattern likely represents systems dominated by riverine runoff. That is, systems in which there is no dominant marine terminating glacier.

When looking at the annual median time series of PAR($0^-$) and $K_{PAR}$ in the shallow zone of each site, it is

possible to see some changes over time (Fig 3). Even though there are very high levels of inter-annual variation of PAR($0^-$) for all sites, the increase seen for Storfjorden is significant (p = 0.01). For $K_{PAR}$, there have been significant increases for Kongsfjorden (p = 0.02) and Porsangerfjorden (p < 0.01). It is important to recall that the monthly climatologies for PAR($0^-$) and $K_{PAR}$ differ (Fig. 2), meaning that changes to one or the other within a given year may result in non-linear changes of $PAR_B$.




**Figure 3:** Annual median values for A) PAR just below the surface (PAR(0⁻)), and B) attenuation coefficient of PAR ($K_{PAR}$) for the shallow pixels (depth ≤ 50 m) from each site. Dashed lines show the linear trend for the values.

### 3.2 Bottom values: $PAR_B$

As $PAR_B$ is available at a monthly resolution (March to October) within this dataset, we can look at how this value has changed across all sites for each individual month (Table 3, Fig. 4). In this way we are able to track changes to the phenology of $PAR_B$. Looking at the median values of $PAR_B$ for all shallow pixels (depth ≤ 50 m), the most notable result is the large change in magnitude between months. June to August generally have much higher $PAR_B$ than March/October, as would be expected. Less expected is the large interannual variance.

This variance may mask significant changes over time. For example, although there is an apparent decrease in $PAR_B$ for the month of June in Kongsfjorden, the change is not statistically significant (simple linear model; $p = 0.4$). The same can be said for the apparent increase in $PAR_B$ for Storfjorden in June ($p = 0.2$). However, there is a significant decrease in $PAR_B$ in Kongsfjorden for September ($p = 0.02$). There is also a significant decrease in $PAR_B$ in Storfjorden for May ($p = 0.04$), July ($p < 0.01$), and August ($p < 0.01$). Porsangerfjorden shows

significant decreases in August ($p < 0.01$) and September ($p < 0.01$). Nuup Kangerlua shows a significant decrease in $PAR_B$ for the month of March ($p = 0.02$).

**Table 3:** Trends for the changes of bottom PAR ($PAR_B$) from 2003 - 2022 for each month (columns) per site (rows). All
units are expressed in rates of mol photons $m^{-2}$ $d^{-1}$ $yr^{-1}$ and are accompanied in brackets by the *p*-value of the fitted linear model. These values therefore show the change in PAR for the given month (columns) per year. Months with significant positive trends are shown in bold, and significant negative trends in bold and italic.

| Site | March | April | May | June | July | August | September | October |
|---|---|---|---|---|---|---|---|---|
| Kongsfjorden | 0 (0.84) | 0.0003 (0.55) | -0.0004 (0.62) | -0.001 (0.44) | -0.0008 (0.34) | -0.001 (0.08) | ***-0.0012 (0.02)*** | *NA* |
| Isfjorden | 0 (0.59) | 0.0008 (0.14) | 0.0013 (0.28) | 0.0017 (0.52) | -0.0015 (0.23) | ***-0.0015 (0.04)*** | -0.0011 (0.18) | *NA* |
| Storfjorden | 0 (0.17) | 0 (0.58) | **0.0007 (0.04)** | 0.0026 (0.25) | ***-0.0039 (< 0.01)*** | ***-0.0011 (0.01)*** | -0.0001 (0.57) | *NA* |
| Young Sound | 0 (0.48) | 0 (0.94) | 0 (0.92) | -0.0004 (0.61) | 0.0001 (0.97) | -0.001 (0.51) | -0.0007 (0.69) | *NA* |
| Qeqertarsuup Tunua | -0.0049 (0.38) | 0.018 (0.39) | -0.0419 (0.53) | -0.0743 (0.21) | -0.015 (0.82) | 0.0066 (0.89) | -0.026 (0.3) | 0.001 (0.86) |
| Nuup Kangerlua | ***-0.023 (0.02)*** | -0.0066 (0.72) | -0.008 (0.75) | -0.0165 (0.65) | 0.0088 (0.76) | -0.007 (0.72) | -0.0113 (0.17) | -0.0061 (0.18) |



| Porsangerfjorden | 0.0001 (0.69) | 0.0001 (0.63) | 0 (0.96) | 0.0006 (0.23) | -0.0001 (0.66) | *-0.0003 (0)* | *-0.0003 (0)* | 0 (0.74) |
|---|---|---|---|---|---|---|---|---|

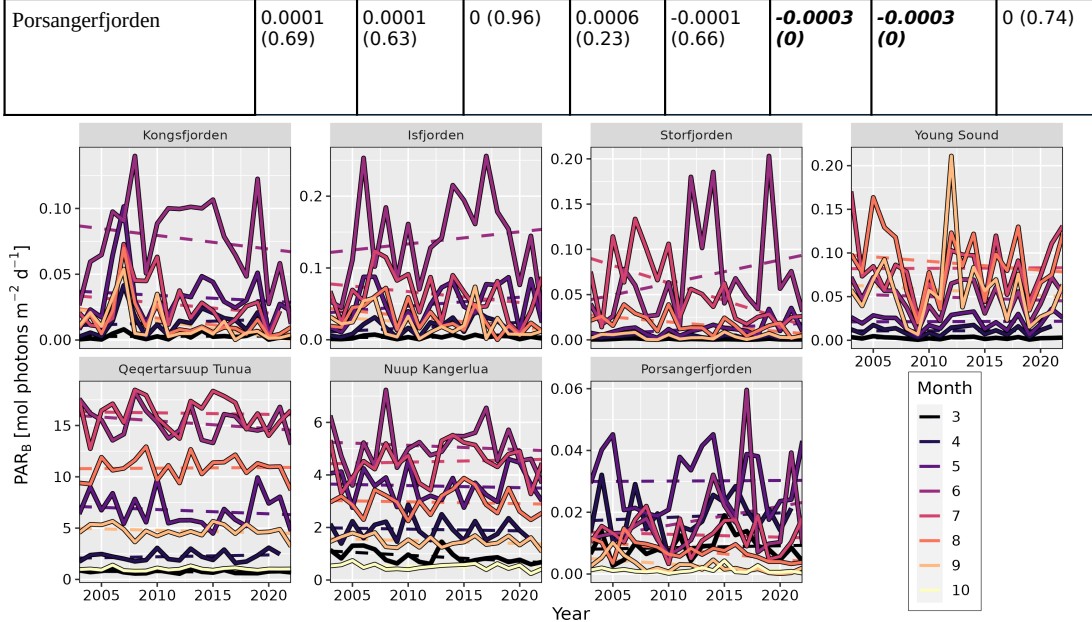

**Figure 4:** Changes in bottom PAR ($PAR_B$) over time by month. Lines represent the median value for all pixels with a depth of 50 m or shallower. Dashed lines show the trend over time, whose slope and *p*-value are given in Table 3. Note the different y-axes between panels.

### 3.3 *P*-functions

The global shallow (i.e. depth ≤ 50 m) *P*-functions show substantial differences between sites (Fig. 5). The shallow seafloor of Western Greenland (Nuup Kangerlua and Qeqertarsuup Tunua) has by far the largest cumulative area receiving the highest levels of $PAR_B$ (25% ≥ 10 mol photons m$^{-2}$ d$^{-1}$), and by far the largest cumulative area receiving ≥ 0.001 mol photons m$^{-2}$ d$^{-1}$ (~90%). This is largely due to the extensive area of open ocean water that is used to estimate $PAR_B$ at these sites. For all other sites, less than 10% of the shallow seafloor receives more than 10 mol photons m$^{-2}$ d$^{-1}$, with ~60% receiving more than 0.001 mol photons m$^{-2}$ d$^{-1}$. Note, however, the difference in the area of the seafloor receiving light levels for Storfjorden (Svalbard) and Porsangerfjorden (Northern Norway). While the percentages for the high and low light levels are similar to most sites, the amounts receiving intermediate light levels are much lower.

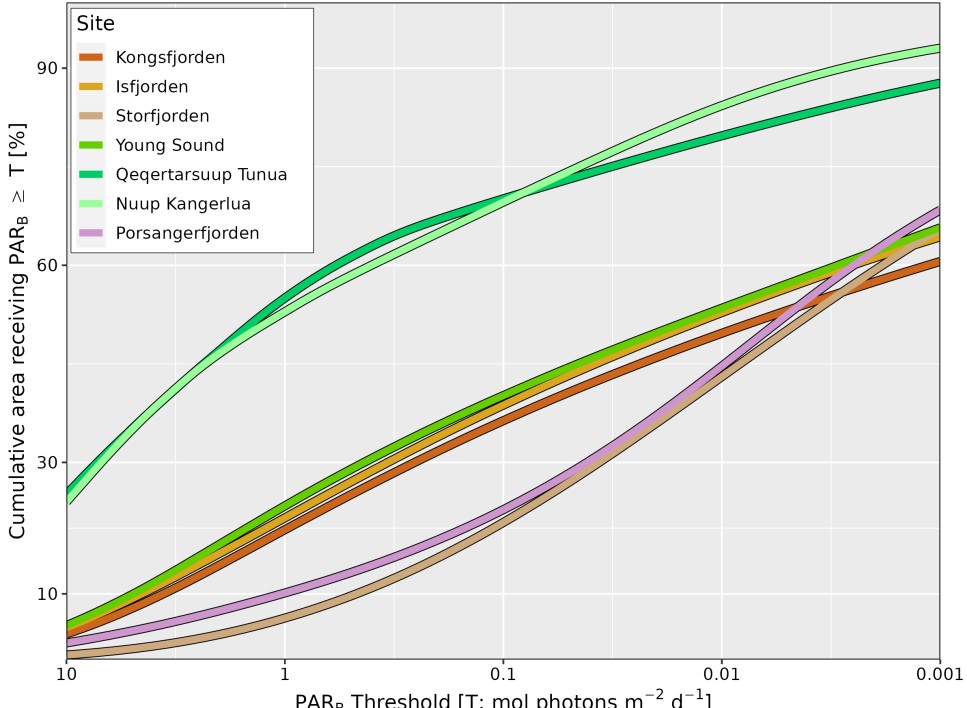

***Figure 5:*** Percent surface area of the seafloor receiving PAR_B above a prescribed threshold (T) at each site. Curves illustrate the global average in percent of the total area (y-axis) of each fjord not deeper than 50 m that experiences the PAR_B value shown on the x-axis. Note that the x-axis is reversed (larger values are on the left) and log10 transformed. For example, Qeqertarsuup Tunua has about 25% of the surface area of the seafloor experiencing a global average of at least 10 mol photons m$^{-2}$ d$^{-1}$, and roughly 90% of the seafloor receiving at least 0.001 mol photons m$^{-2}$ d$^{-1}$.

Within these sites, there are also different patterns in the monthly climatology (Fig. S1). Generally the peak in PAR_B for all sites, at both high and low levels of PAR_B, occurs in June, with a build up to (and down from) this peak over the preceding (following) three to four months. Exceptions to this pattern may be seen in Young Sound and Qeqertarsuup Tunua where the peak months of PAR_B occur between July and August. These patterns are driven by the combined effect of many variables: solar zenith angle, cloud and ice cover, $K_{PAR}$, and the underwater fjord morphology.

Using the annual P-functions per site, we may see that the shallow areas receiving high levels of PAR_B across all sites have not changed much from 2003 to 2022 (Fig. S2). The shallow areas receiving lower levels of PAR_B for Western Greenland (Qeqertarsuup Tunua and Nuup Kangerlua) have also remained relatively stable. Increasing interannual variance in the *P*-functions for low light may be seen in Eastern Greenland (Young Sound), Northern Norway (Porsangerfjorden), and Western Svalbard (Kongsfjorden and Isfjorden) respectively (Fig. S2).





Changes in PAR($0^-$), $K_{PAR}$, and $PAR_B$ are interesting on their own, but when used in combination with known photic limits for ecologically important species, the results become illuminating.

**3.4 Changes to inhabitable benthic area for kelp growth**

In the interest of demonstrating a clear use case for this dataset, the biologically relevant PAR limitation of 47 mol photons $m^{-2}$ $y^{-1}$ (median of Table 2) was converted to 0.13 mol photons $m^{-2}$ $d^{-1}$, by simply dividing 47 by 365, and used as a filter to investigate changes to the shallow bottom area within fjords where ecologically important species (kelp) could survive (Table 4). By utilising the annual $PAR_B$ data, we were able to see what

percentage of the shallow area of each fjord should be able to support benthic macroalgal communities, and if any changes have occurred over time. One may see that the annual spatial area changes somewhat between years, but a significant decrease is only seen in Kongsfjorden.

**Table 4:** Area of the fjords capable of supporting benthic macroalgae. The total shallow area (≤ 50 m; $km^2$) of each site is shown, followed by the global suitable area (% of shallow area averaged over the full dataset). The year at which the lowest and highest values for spatial availability were observed. The linear trend (% spatial availability / year; $p$-value) in the dataset from 2003-2022 is also provided. Note that this is the trend value for the full time series, not the trend between the high and low columns also provided in this table. Significant negative trends are shown in bold and italic. Note that the trend values are in percent values, meaning a slope of -0.21% would mean a reduction of 2.1% of available substrate over 10

years.

| Site | Total shallow area [$km^2$] | Global average [%] | Lowest [% (year)] | Highest [% (year)] | Trend [%/y ($p$-value)] |
|---|---|---|---|---|---|
| Kongsfjorden | 106 | 41% | 32% (2020) | 44% (2007) | ***-0.21 (0.04)*** |
| Isfjorden | 774 | 45% | 35% (2020) | 44% (2007) | -0.08 (0.48) |
| Storfjorden | 2,770 | 27% | 19% (2011) | 32% (2019) | -0.08 (0.60) |
| Young Sound | 104 | 43% | 31% (2009) | 42% (2012) | -0.13 (0.33) |
| Qeqertarsuup Tunua | 3,493 | 69% | 64% (2013) | 68% (2013) | 0.01 (0.83) |
| Nuup Kangerlua | 1,006 | 67% | 57% (2009) | 63% (2017) | 0.00 (0.98) |
| Porsangerfjorden | 337 | 25% | 20% (2010) | 26% (2017) | -0.05 (0.46) |

**4 Code and data availability**

The code written for the analysis of these data, and the creation of the figures, may be found on GitHub at: https://github.com/FACE-IT-project/fjord_PAR (last access: 13 November 2023).

The PAR dataset may most easily be accessed via the R package 'FjordLight', which can be installed via CRAN (Gentili et al., 2023b) or GitHub at: https://face-it-project.github.io/FjordLight (last access: 13 November 2023). The data are also available for download at the World Data Center PANGAEA as a series of NetCDF files, one for each fjord: https://doi.pangaea.de/10.1594/PANGAEA.962895



All data were generated from a base of daily gridded remotely sensed observations (see Sections 2.2 and 2.3). The primary variables created are: PAR(0⁻), $K_{PAR}$, and $PAR_B$ (Table 5). These three variables are available at ~ 50m resolution across all sites. P-functions, which are a summary value and therefore not gridded, were computed from $PAR_B$ and the surface area of all shallow (depth ≤ 50 m) and coastal (depth ≤ 200 m) pixels. For these four primary variables, four different levels were created and are available in this dataset:

• *Monthly:* The average of all available daily data within a given month. Expressed as units of mol photons m-2 day-1 (except $K_{PAR}$ [m-1]). For example, the monthly value for June 2006 is the average of all available days of data from 2006-06-01 to 2006-06-30. **NB:** A given pixel during a given month was required to have at least 20 days of available data to be included in the dataset.

• *Climatology:* The average of all of the same months of data across the available years of data. For
example, the July climatological value is the average of all July monthly values from 2003 to 2022. **NB:** Due to latitudinal differences, some sites do not receive 20 days of light in October, and so are missing October climatologies (e.g. Kongsfjorden).

• *Yearly:* The average of all available monthly values during a given year.

• *Global:* The average of all yearly values.

Note that because the primary utility of this dataset is identified to be $PAR_B$, only these data are available at the monthly temporal resolution. This technical choice was made because the inclusion of all monthly data for all variables would make the NetCDF files too large to load into memory for anything other than servers or very powerful desktop computers. It was determined that this would severely limit the usability of these data, and therefore it was preferable to remove the monthly data for PAR(0⁻), $K_{PAR}$, and the P-functions. The NetCDF files
contain a range of meta-data that may also be of interest to users (Table 6).

**Table 5:** The code names (rows) for the available data (columns) for the PAR values provided in the dataset.

| Variable | Global value | Annual value | Climatology value | Monthly value |
|---|---|---|---|---|
| P-function Coastal (≤ 50 m) | GlobalPcoastal | YearlyPcoastal | ClimPcoastal | NA |
| P-function Shallow (≤ 200 m) | GlobalPshallow | YearlyPshallow | ClimPshallow | NA |
| PAR(0⁻) | GlobalPAR0m | YearlyPAR0m | ClimPAR0m | NA |
| $K_{PAR}$ | GlobalKpar | YearlyKpar | ClimKpar | NA |
| $PAR_B$ | GlobalPARbottom | YearlyPARbottom | ClimPARbottom | MonthlyPARbottom |

**Table 6:** Secondary variables of interest provided within the dataset

| Variable | Definition |
|---|---|
| name | Short code name assigned to each study site. Used within the code for the R package 'FjordLight'. |
| longitude/latitude | The coordinates of a given pixel in decimal degrees (EPSG:4326 projection). |
| Months | The months available within the dataset in integers (i.e. 3 - 10 for March - |





| | October). |
|---|---|
| Years | The years available in the dataset expressed as integers (i.e. 2003 - 2022). |
| irradianceLevel | The values (mol m$^{-2}$ day$^{-1}$) used to define the steps (x-axis) in the P-functions. |
| depth | Depth expressed as negative values [m] |
| elevation | Elevation above sea level expressed as positive values [m] |
| area | Surface area of pixel [km$^2$] |
| AreaOfCoastalZone | Sum of the surface area (km$^2$) of the pixels within the study site with a depth of 200 m or shallower. |
| AreaOfShallowZone | Sum of the surface area (km$^2$) of the pixels within the study site with a depth of 50 m or shallower. |
| site_average_longitude/latitude | The central coordinates of the site in decimal degrees (EPSG:4326 projection). |


## 5 Conclusion

The data product summarised in this report was designed to provide a number of variables throughout a range of EU Arctic fjords. Monthly PAR$_B$ is the primary variable of interest, but the dataset also provides global mean values, annual values, and monthly climatologies for: PAR(0$^-$), $K_{PAR}$, and PAR$_B$. The data are currently available

from 2003 to 2022, but could potentially be updated annually because they are created from algorithms that utilise operational data streams applicable to MODIS, but also to VIIRS and future PACE missions.

With the exception of Kongsfjorden (Svalbard), the available PAR$_B$ in the EU Arctic fjords shows no significant signs of long-term change. However, although PAR(0$^-$) and $K_{PAR}$ are not changing much at an annual

rate, PAR$_B$ values of certain months (e.g. July) are changing more rapidly than others. It is also important to note the large inter- and intra-annual variance seen in the various PAR measurements. In some cases, there are strong upward trends in PAR$_B$ during the early and late months of the year, which is interpreted here as an extension of the sea-ice free period. As sea-ice melts earlier, and freezes later, more light reaches the bottom during the months that historically have had lower light levels. However, there is still an overall (not statistically

significant) downward trend in the annual averages due to increased light attenuation in the water column ($K_{PAR}$) during the peak months of the year. This reduction in PAR$_B$ is interpreted to be caused by increased terrestrial/glacial runoff into fjords, which causes darkening of the water due to the delivery of dissolved and particulate material. The darkening effect has an overall larger impact on the annual PAR$_B$, because its timing coincides with the peak in PAR(0$^-$). These results are consistent with those of Singh et al. (2022) at Pan-Arctic

scale, but emphasise the need to consider the local processes that control light attenuation.

Because benthic algae rely on photosynthesis to survive throughout the year, any reduction in PAR$_B$ at water depths where the algae are light limited is likely to have a negative effect. However, the changes observed in the PAR dataset are complex, and clearly non-linear. Therefore, trends presented here should not be

extrapolated into the future. It is also known that at some point in the future the peak rates of terrestrial/glacial





runoff will be reached (this may have already occurred in parts of Svalbard), after which fjord waters are expected to lighten again as terrestrial/glacial runoff reduces. One must also consider that PAR thresholds for important benthic species are to some extent driven by ambient seawater temperature (e.g. eelgrass - Staehr & Borum, 2011; kelps - Niedzwiedz & Bischof, 2023). Therefore, as the Arctic warms, PAR thresholds (and thus

historical depth ranges) will change regardless of how turbid the water may or may not be in the future.

As shown in the inhabitable area example (Section 3.4), this dataset can be useful for a suite of research questions (e.g., Fig 1). The high spatial resolution $PAR(0^-)$, $K_{PAR}$, and $PAR_B$ values can also be integrated into a workflow that uses any number of other datasets. For example, species distribution modelling (SDM) within

fjords must be done at very high resolution, but tends to use only global values. Whereas the life cycle of an organism within the water column could be better understood by utilising the monthly climatologies of $K_{PAR}$. Or the use of annual $PAR(0^-)$ to understand changes in irradiance received by fjord surface waters over time. Other examples include the potential benefits of using this dataset for forcing or initial conditions in state-of-the-art ocean-biogeochemical models and/or forcing for ecosystem box models. The purpose here is to demonstrate

some of the many potential applications of this dataset, which fills a gap in the physical understanding of EU Arctic fjord systems.

**Author contributions**

The paper was conceived by JPG and the original dataset created by BG, based on prior work done by BG, JPG, RKS, and SB. The analyses and visualisations were performed by RWS. All authors contributed to the writing

and editing of the manuscript. RWS coordinated the editing and managed the submission.

**Competing interests**

The authors declare that they have no conflict of interest.

**Acknowledgements**

This study is a contribution to the project FACE-IT (The Future of Arctic Coastal Ecosystems – Identifying

Transitions in Fjord Systems and Adjacent Coastal Areas).

**Financial support**

FACE-IT has received funding from the European Union's Horizon 2020 research and innovation programme under grant agreement no. 869154.





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
