# Peer review of "Underwater light environment in Arctic fjords"

_Earth System Science Data, 2023_

## Referee Comment (RC1)

The authors combined and processed data from different long-term satellite data-sets, along with high resolution bathymetry, to estimate PAR, $K_{PAR}$ and $PAR_B$ in six fjords in the Arctic Ocean. The aim of this work is interesting as these data can support investigations about climate changes in the region. Nevertheless, the satellite-derived data-set is strongly related with the environmental characteristics of the water column, but any in situ observation is available to assess the quality and reliability of their results. The use of minimum light requirement is a very poor and qualitative indication. The added value and the effort they did is also in the geographical selection to obtain the data for each of the fjords, characterized by a complex topography, but this is not properly described. The data-set can be better documented and even the methods and the statistics applied is questionable. That's why I suggest a major revision of the paper.

Specific comments

-Maps of the fjords reporting horizontal scale, bathymetry and position of pixels can be added to Fig.1. You could also indicate here how many pixels were available and the surface of shallow and coastal surface (which is only reported at the end of the paper)

- Despite some information are spread in the text, a table should resume detailed information about the data / sensors used, along with period covered, temporal and spatial coverage.

- Even you mention that only pixel with a minimum of 20 values each month were considered, you should provide some statistics about the temporal distribution of good / discarded data at least for each year and each fjord.

-In Fig.2 use different colours to indicate each fjord: C D and E F can be hardly distinguished in the reported plots.

- The resolution of the computed data-set is at 50 m but satellite data are at 1 km. This may result into misleading interpretation for other users and should be clearly indicated in the text, along with the method used for interpolation.

-Reported climatological averages need the standard deviations, otherwise some results are meaningful

-Can you explain why median better describe the seasonal cycle?

-I would avoid estimation of long trend as the derived data-set are semi-qualitative and values are strongly dependent of the ice formation/ melting cycle. On the contrary you could better relate and discuss the observed interannual variability in terms of sea-ice, cloud coverage and river runoff, but this might lie outside the scope of this Journal

---

## Referee Comment (RC2)

Specific comments:

Page 7 line 154: The fjords of Northen Norway …. Looking at figure 1 and Table 1 there is only 1 fjord in Norway.

Page 15 , 3.2 Bottom values…. The description of table 3 and figure 4 has no additional information than describing the values. Is there a way to provide a clear message to the reader on the meaning of these values?

Pag. 17 line 364: which is figure S1?

Pag. 17 line 372: same for figure S2

---

## Author Response (AR1)

**Reviewer #1 Response**

We thank the reviewer for their thoughtful feedback on this manuscript. Below we provide responses to each point raised.

The authors combined and processed data from different long-term satellite data-sets, along with high resolution bathymetry, to estimate PAR, KPAR and PARB in six fjords in the Arctic Ocean. The aim of this work is interesting as these data can support investigations about climate changes in the region. Nevertheless, the satellite-derived data-set is strongly related with the environmental characteristics of the water column, but any in situ observation is available to assess the quality and reliability of their results.

- Unfortunately there is an extreme paucity of *in situ* measured PAR data in the Arctic. It is specifically for this reason that the dataset described in this manuscript was created. As is explained in the introduction. For this reason we have not validated this dataset against *in situ* measurements. Which we assume is what the reviewer is alluding to with their comment. It should also be noted that Singh et al. (2022), which is referenced throughout this manuscript, reported on the uncertainties in both *in situ* and remotely sensed PAR values in the Arctic.

The use of minimum light requirement is a very poor and qualitative indication.

- We disagree that minimum light requirements is a poor indicator for many photosynthetic species, nor is it a qualitative value. Rather it is a quantitative and ecologically relevant parameter (see Gattuso et al., 2006 for a review).

The added value and the effort they did is also in the geographical selection to obtain the data for each of the fjords, characterized by a complex topography, but this is not properly described.

- We fail to understand this comment as the bathymetry datasets describing the topography of the sites are fully documented in section 2.2.1.

The data-set can be better documented and even the methods and the statistics applied is questionable.

- The reviewer notes that the applied statistics are questionable, but only mentions the use of median values in their specific comments, so we assume their criticism is limited to this one statistical choice. Median values are used to report the spatial averages throughout the manuscript due to the kurtosis (right skew) of the data. This is caused by some very shallow pixels having very high PAR values that unduly influence the mean value for the study sites.

Specific comments
-Maps of the fjords reporting horizontal scale, bathymetry and position of pixels can be added to Fig.1. You could also indicate here how many pixels were available and the surface of shallow and coastal surface (which is only reported at the end of the paper)

- Agreed. Figure 1 has been overhauled. Horizontal scales (lon/lat) have been added. In this figure and all others, the colour palette for the sites has been changed to one that shows much clearer contrast.
- A new table has also been added after the figure to show how many shallow and coastal pixels are available per site.

- Despite some information are spread in the text, a table should resume detailed information about the data /sensors used, along with period covered, temporal and spatial coverage.

- We appreciate that such a table may make this information more readily available to the reader, but we have opted not to add another table as there are already numerous. Rather we

have ensured that the information requested by the reviewer is provided in the text, making it easier to find when scanning the section on the satellite products.

- Even you mention that only pixel with a minimum of 20 values each month were considered, you should provide some statistics about the temporal distribution of good / discarded data at least for each year and each fjord.

- Done. A new table (Table S1) has been provided that shows the percent of pixels per site, per month, per year that were removed.

-In Fig.2 use different colours to indicate each fjord: C D and E F can be hardly distinguished in the reported plots.

- Agreed. The colour palette for the sites has been changed for all relevant figures.

- The resolution of the computed data-set is at 50 m but satellite data are at 1 km. This may result into misleading interpretation for other users and should be clearly indicated in the text, along with the method used for interpolation.

- Agreed. The method of downscaling from 1 km to ~50 - 200 m has been more clearly explained in a new paragraph at the end of Section 2.3.

- Reported climatological averages need the standard deviations, otherwise some results are meaningful

- Done. The standard deviations for all monthly and yearly climatology values were re-run and packaged in a set of addendum NetCDF files which have been published on PANGAEA. Shaded ribbons have also been added to Figures 2 and 3 to show the SD for the annual mean and monthly climatology values.

-Can you explain why median better describe the seasonal cycle?

- Done. Median values are preferable to mean when describing the seasonal cycle due to the kurtosis (right skew) of the PAR variables. This statistical choice has been made more clear in the text, starting with Section 3.1 P1.

-I would avoid estimation of long trend as the derived data-set are semi-qualitative and values are strongly dependent of the ice formation/ melting cycle. On the contrary you could better relate and discuss the observed interannual variability in terms of sea-ice, cloud coverage and river runoff, but this might lie outside the scope of this Journal

- We respectfully disagree that this PAR dataset is 'semi-qualitative'. It is definitely quantitative. We do however agree that the extrapolation of a 20 year time series to a long-term trend should be done with caution (if at all), and that the inter-annual variability is likely more related to the physical phenomenon mentioned in the reviewers comment. We have reiterated this point in the second paragraph of the Conclusion and have ensured that we remind readers of the pitfalls of extrapolating from such a short period with so much inter-annual variability.

References for this response:

Singh, R. K., Vader, A., Mundy, C. J., Søreide, J. E., Iken, K., Dunton, K. H., Castro de la Guardia, L., Sejr, M. K., and Bélanger, S.: Satellite-Derived Photosynthetically Available Radiation at the Coastal Arctic Seafloor, Remote Sens., 14, 5180, https://doi.org/10.3390/rs14205180, 2022.

Gattuso J.-P., Gentili B., Duarte C. M., Kleypas J. A., Middelburg J. J. & Antoine D., 2006. Light availability in the coastal ocean: impact on the distribution of benthic photosynthetic organisms and their contribution to primary production. Biogeosciences 3:489-513. https://doi.org/10.5194/bg-3-489-2006

**Reviewer #2 Response**

We thank the reviewer for their thoughtful feedback on this manuscript. Below we provide responses to each point raised.

The paper is well written and the information well organised even if it's not clear what are the complementary information/data/observation required by this methodology. A key point is missing, the uncertainty associated to the explained methodology and the accuracy of the information of this dataset.

- We think that the required data sources and methodological implementation are well outlined in Sections 2.2.2 and 2.3. We also think that providing an in depth technical description of this methodology is outside of the scope of this ESSD dataset paper. As discussed in the manuscript, the interested reader can see Singh et al. (2022) for more detail. The same paper provides in-depth analyses of the uncertainties involved.
- This applies to the following point as well.

There is no comparison with any other methodology that can be used in this field, is this the only way for estimating gridded PAR in Arctic areas? A critical description of the chosen methodology, its limitations and how it performs compare to others is missing.

Yes, this methodology is the only one we are aware of that allows for the large-scale, high-resolution, long-term estimates of PAR in coastal Arctic waters.

An interesting point would be to explain the sustainability of this methodology for updating the existing this timeseries.

- We agree with the author and have added two sentences to the first paragraph of the conclusion section discussing this point.

Specific comment:

Page 7 line 154: The fjords of Northern Norway …. Looking at figure 1 and Table 1 there is only 1 fjord in Norway.

- Generally speaking, Porsangerfjorden is considered to be representative of Northern Norwegian fjords. We have added a sentence to the end of Section 2.1 Paragraph 2 clarifying this point.

Page 15 , 3.2 Bottom values…. The description of table 3 and figure 4 has no additional information than describing the values. Is there a way to provide a clear message to the reader on the meaning of these values?

- Because Section 3.2 is meant to report on the results, and their context is discussed in detail in Section 5 Paragraph 2, we do not want to repeat to much text here. We have added a sentence at the end of Section 3.2 Paragraph 1 giving the reader the context of why the numbers are important.

Page. 17 line 364: which is figure S1?

- Perhaps the supplementary figures were not made available upon the initial upload. They are attached here and below and will be in the final version of the manuscript.

[Figure]

**Figure S1:** Shallow (i.e. depth ≤ 50 m) P-functions per site for the monthly climatology of bottom PAR (PAR_B). Otherwise the same as Fig. 5.

Pag. 17 line 372: same for figure S2

[Figure]

**Figure S2:** Shallow (i.e. depth <= 50 m) P-functions per site for annual average bottom PAR (PAR$_B$). Otherwise the same as Fig. 5.

References in this response:

Singh, R. K., Vader, A., Mundy, C. J., Søreide, J. E., Iken, K., Dunton, K. H., Castro de la Guardia, L., Sejr, M. K., and Bélanger, S.: Satellite-Derived Photosynthetically Available Radiation at the Coastal Arctic Seafloor, Remote Sens., 14, 5180, https://doi.org/10.3390/rs14205180, 2022.